# Segregation of pathways leading to pexophagy

Francesco G Barone , Sylvie Urbé , Michael J Clague

**Peroxisomes are organelles with key roles in metabolism including long-chain fatty acid production. Their metabolic functions overlap and interconnect with those of mitochondria, with which they share an overlapping but distinct proteome. Both organelles are degraded by selective autophagy processes termed pexophagy and mitophagy. Although mitophagy has received intense attention, the pathways linked to pexophagy and associated tools are less well developed. We have identified the neddylation inhibitor MLN4924 as a potent activator of pexophagy and show that this is mediated by the HIF1α-dependent up-regulation of BNIP3L/NIX, a known adaptor for mitophagy. We show that this pathway is distinct from pexophagy induced by the USP30 deubiquitylase inhibitor CMPD-39, for which we identify the adaptor NBR1 as a central player. Our work suggests a level of complexity to the regulation of peroxisome turnover that includes the capacity to coordinate with mitophagy, via NIX, which acts as a rheostat for both processes.**

## Introduction

Selective autophagy requires that an organelle destined for elimination is marked and then linked to the phagophore membrane. This normally occurs via a LC3-interacting region (LIR) that links to lipid-modified LC3 (1, 2). In many cases, these selective autophagy adaptors are recruited to damaged organelles by binding to ubiquitin, which accumulates on their surface. These include adaptors from the sequestosome-1–like receptor family that include p62, NBR1, NDP52, TAX1BP1, and OPTN. All of these proteins have both a LIR and a ubiquitin-binding domain. Nevertheless, they are differentially associated with various forms of selective autophagy, and the underlying reasons are not completely understood (3). A comparison of these factors in the regulation of ubiquitin-dependent mitophagy suggested that only NDP52 and OPTN play critical roles (4). In contrast, ubiquitin-dependent pexophagy relies upon NBR1 (5, 6). There is also diversity of mechanisms in pathways converging on organelle turnover. For example, mitophagy is induced by mitochondrial depolarisation, activating the PINK1-PRKN pathway which leads to

coating the outer mitochondrial surface with ubiquitin and is suppressed by the deubiquitylase (DUB), USP30 (7, 8). However, the vast majority of mitophagy in an organism is PINK1-PRKN independent (9, 10). An alternative ubiquitin-independent pathway exists, which employs trans-membrane domain–containing proteins BNIP3 and BNIP3L/NIX, which insert directly into the polarised mitochondrial membrane and link to the phagophore LC3 by LIR domains (11). NIX plays a critical role in the removal of mitochondria during reticulocyte development and neuronal differentiation and is up-regulated by hypoxia (12, 13, 14, 15, 16).

Inherited mutations in peroxisomal genes can lead to debilitating peroxisomal disorders. Many of these have been linked to peroxisome biogenesis, but it is now apparent that the other arm of peroxisome homeostasis, namely pexophagy, is also critical (17, 18). For example, the AAA-ATPase comprising PEX1, PEX6, and PEX26 is a pexophagy suppressor by virtue of removing the ubiquitylated peroxisomal matrix protein import receptor Ub-PEX5 from the peroxisomal membrane (19). This shift into the spotlight has highlighted an unmet need for tools to manipulate pexophagy in a controlled manner. Here, we started out by surveying a number of candidate compounds and discovered that the neddylation inhibitor MLN4924 and the USP30 inhibitor compound 39 (CMPD-39) are both potent inducers of pexophagy (20, 21). In parallel to mitophagy, we show that the MLN4924 effect reflects activation of HIF1α and up-regulation of NIX. In contrast, CMPD-39 promotes a pexophagy pathway that uses the ubiquitin-recognising adaptor NBR1, consistent with the proposed mode of action on a DUB previously linked to pexophagy (22, 23). Our results highlight the increasing awareness of the complexity of pexophagy pathways akin to mitophagy and suggest that, in the case of hypoxia/MLN4924 treatment, both organelles can be turned over in a coordinated manner.

## Results

### Survey of pexophagy-inducing agents

We generated a retinal pigment epithelial (hTERT-RPE1) cell line, expressing a peroxisomal matrix targeting signal joined with a mKeima fluorophore (Keima-SKL). This fluorophore reports on lysosomal delivery of peroxisomes. The lower pH of the lysosome

Molecular Physiology and Cell Signalling, Institute of Systems, Molecular and Integrative Biology, University of Liverpool, Liverpool, UK

Correspondence: clague@liv.ac.uk; urbe@liv.ac.uk

leads to a change in the excitation spectrum properties of the reporter and the resultant pexolysosomes are represented in red pseudocolour (Fig 1A) (22, 24). We conducted a survey of various chemicals, drawn from the literature, for their influence on pexophagy. These included agents linked to peroxisome turnover (4-phenylbutyric acid (4-PBA), Clofibrate), others previously linked to mitophagy (deferiprone (DFP), MLN4924), the USP30 inhibitor CMPD-39, which we have previously shown to promote both pathways as well as a general oxidative stress ($H_2O_2$) (21, 25, 26, 27).

All agents led to increases in pexophagy indices of similar magnitudes (Figs 1B–D and S1A–C). Both DFP and MLN4924 generate an increase in HIF1α, which in turn drives the expression of the mitophagy adaptors BNIP3 and NIX (Fig 1E–H). DFP inhibits the prolyl hydroxylase enzyme that renders HIF1α a substrate for the Cullin RING ligase VHL, whereas MLN4924 inhibits the neddylation-dependent activation of all Cullins (Fig 1E). We have subsequently focused our efforts on characterising CMPD-39 and MLN4924 because of their high selectivity for their protein targets.

### Elevated NIX levels promote pexophagy

We next showed that MLN4924-induced pexophagy requires the canonical autophagy machinery as the number of pexolysosomes is sensitive to depletion of the key autophagy orchestrator ATG7 (Fig 2A–D). It is insensitive to depletion of the peroxisomal membrane protein ACBD5, whose yeast counterpart Atg37 has been strongly linked to pexophagy (28). However, the combined depletion of BNIP3 and NIX restores pexophagy to baseline levels, suggesting that these proteins fully account for the observed effect of MLN4924 (Fig 2A–D). Individual depletions of BNIP3 and NIX indicate that NIX is the principal inducer of pexophagy (Fig 2E–H).

### Direct association of NIX with peroxisomes

Although NIX is known to directly insert into the mitochondrial membrane (Fig S2), its association with peroxisomes has not been previously established. We wished to check if we could observe NIX on peroxisomes and secondly, whether this reflects a direct association or transfer from mitochondria. We employed a strategy we have previously used to demonstrate insertion of USP30 directly into the peroxisomal membrane (22). We elected to effectively remove mitochondria from hTERT-RPE1 cells, stably expressing high levels of YFP-Parkin, by eliciting mitophagy after mitochondrial depolarisation with antimycin and oligomycin (A/O, Fig 3A). Subsequently, cells were transfected with DsRed-NIX and examined by fluorescence microscopy for colocalisation with the peroxisomal marker PMP70 (Fig 3B and C) or harvested for subcellular fractionation (Fig 3D). We found clear evidence for direct association of NIX with peroxisomal membranes under these conditions, suggesting that the effect of NIX on pexophagy reflects this association rather than some indirect consequence of mitophagy. After the depletion of mitochondria, a significantly higher fraction of residual endogenous NIX is recovered in the light membrane fraction containing peroxisomes (Fig 3D, indicated by black and red arrowheads).

### CMPD-39–induced pexophagy requires NBR1

NIX expression is under the control of ubiquitin E3-ligase activity through regulation of expression and stability by VHL and FBXL4 respectively, both of which are inhibited by MLN4924 (29 Preprint). However, the actual pexophagy event does not require organelle coating with ubiquitin, owing to its direct insertion into membranes. Moreover, CMPD-39 does not increase NIX levels (Fig 1E and G). We presumed that USP30 inhibition by CMPD-39 must be promoting pexophagy via an alternative ubiquitin-dependent pathway. We thus chose to test a role, in this system, for the established pexophagy adaptor NBR1, which acts as a bridge between ubiquitin and LC3. The increased pexophagy after CMPD-39 is insensitive to BNIP3/NIX or ACBD5 depletion but returns to baseline on depletion of ATG7 or NBR1 (Fig 4A–D).

## Discussion

Here, we have used MLN4924 to induce NIX via the HIF1α pathway (Fig 4E). We show that NIX is targeted to peroxisomes and promotes pexophagy. This provides an additional function for NIX, beyond its established role in governing mitophagy in various contexts (12, 14, 15, 16, 30). Thus, NIX-dependent pexophagy likely contributes to an adaptive metabolic reprogramming in response to hypoxia (31). While this article was in preparation, Ganley and colleagues reported similar findings, using the iron-chelating drug DFP to induce HIF1α and hence NIX (32). In our hands, DFP has a stronger pexophagy-promoting effect than MLN4924 despite the latter's complete inhibition of neddylation. As the CRL2^VHL complex is the principal regulator of HIF1α stability and requires neddylation for activity, we suggest that there may be additional ill-defined pathways induced by DFP (33). Both drugs provide a complementary approach to inducing pexophagy and can be usefully cross-referenced with each other in future studies. MLN4924 (otherwise known as pevonedistat) is well tolerated and has featured in more than 30 oncology-related clinical trials (34). In this disease context, up-regulation of the HIF1α pathway is not desirable and our work highlights that consideration should be given to combination therapy together with HIF pathway inhibitors.

It has been estimated that 2/3 of the respective proteomes of mitochondria and peroxisomes overlap (35). There are close contacts between the two organelles and some mitochondrial material can be delivered to peroxisomes via vesicular transport (36). We have previously shown that USP30 can be directly targeted to peroxisomes in cells lacking mitochondria (22). We now show that the same holds true for NIX. Thus, the effect of NIX on pexophagy does not require its traversal through mitochondria.

We have also previously shown that CMPD-39 induces pexophagy (21). Here, we have been able to compare the magnitude of this effect with other agents. In our survey of pexophagy-inducing chemicals, CMPD-39–induced pexophagy to a similar degree as MLN4924. However, CMPD-39–induced pexophagy is mostly dependent on NBR1 and this distinguishes it from the MLN4924-NIX pathway (Fig 4E). The ubiquitin-binding pexophagy adaptor NBR1 has previously been shown to be necessary and sufficient for basal

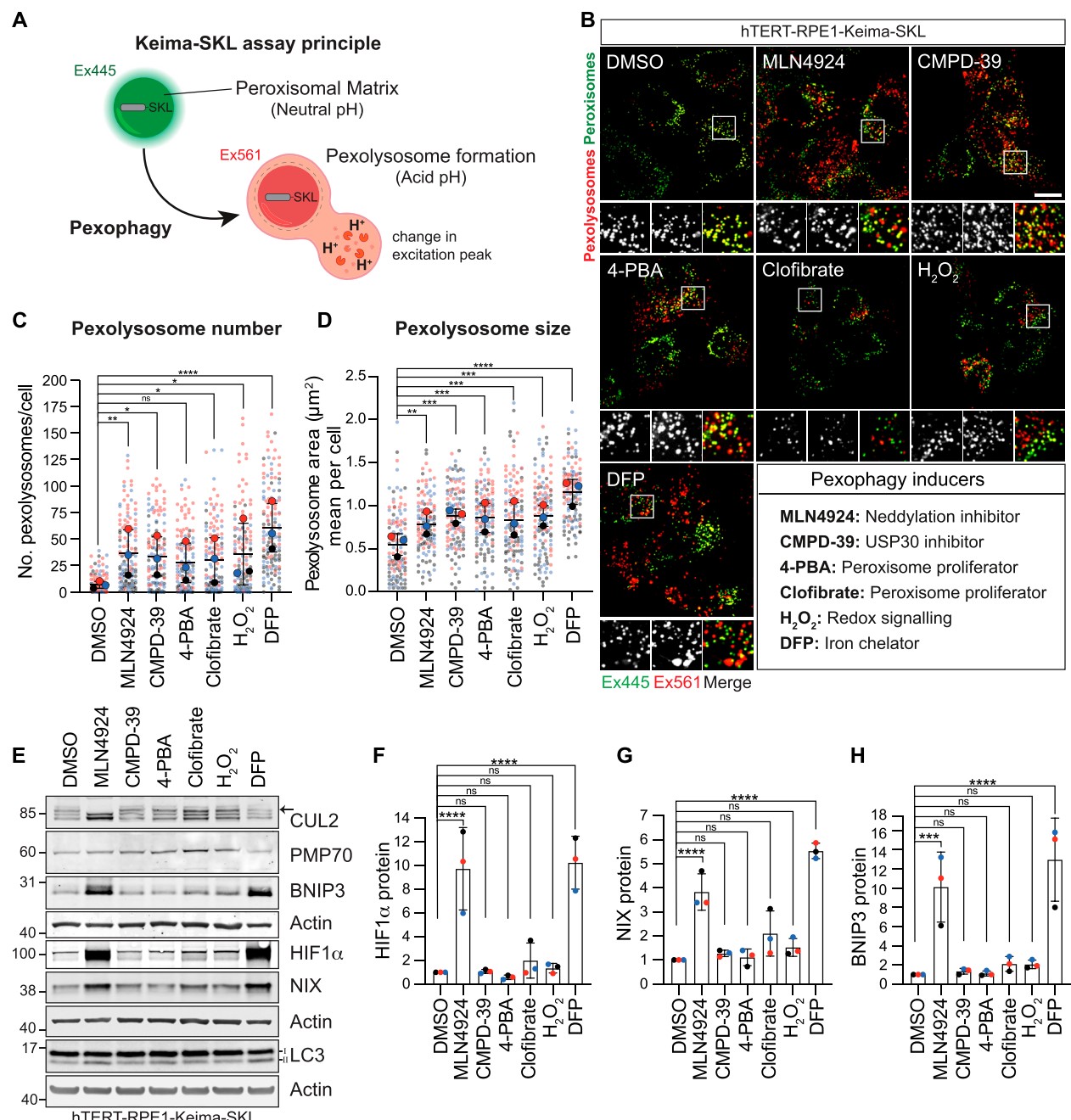

**Figure 1. A chemical screen for pexophagy inducers.**
**(A)** Schematic for the Keima-SKL pexophagy reporter system. The Keima fluorescent reporter is targeted to the peroxisomal matrix via the peroxisomal-targeting signal 1 (tripeptide SKL). Upon delivery to the acidic environment of lysosomes, the excitation spectrum of Keima is red shifted. Fluorescence emissions from these two excitation wavelengths (445 and 561 nm) are pseudocoloured green and red, respectively. **(B)** Representative images of hTERT-RPE1 cells stably expressing the Keima-SKL pexophagy reporter. CMPD-39 (1 μM) was administered for 96 h before imaging. For all other conditions, cells were treated for 24 h with MLN4924 (1 μM), 4-PBA (4-PBA, 1 mM), clofibrate (20 μM), hydrogen peroxide ($H_2O_2$, 100 μM), and deferiprone (DFP, 1 mM). Scale bar 20 μm. **(C, D)** Graphs illustrate the number and mean area of pexolysosomes per cell. Quantification of the data from three independent colour-coded experiments is shown. Mean and SD are indicated; >40 cells were quantified per condition in each replicate experiment. One-way ANOVA and Bonferroni's multiple comparison test. *P < 0.05, **P < 0.01, ***P < 0.001, ****P < 0.0001. **(E)** Representative Western blot of hTERT-RPE1-Keima-SKL treated as in (B), probed for Cullin-2 (CUL2), PMP70, BNIP3, HIF1α, LC3, NIX, and actin. Arrow indicates the neddylated form of Cullin-2. **(F, G, H)** Quantitation of data shown in (E), indicating the mean and SD for three independent colour-coded experiments.
Source data are available for this figure.

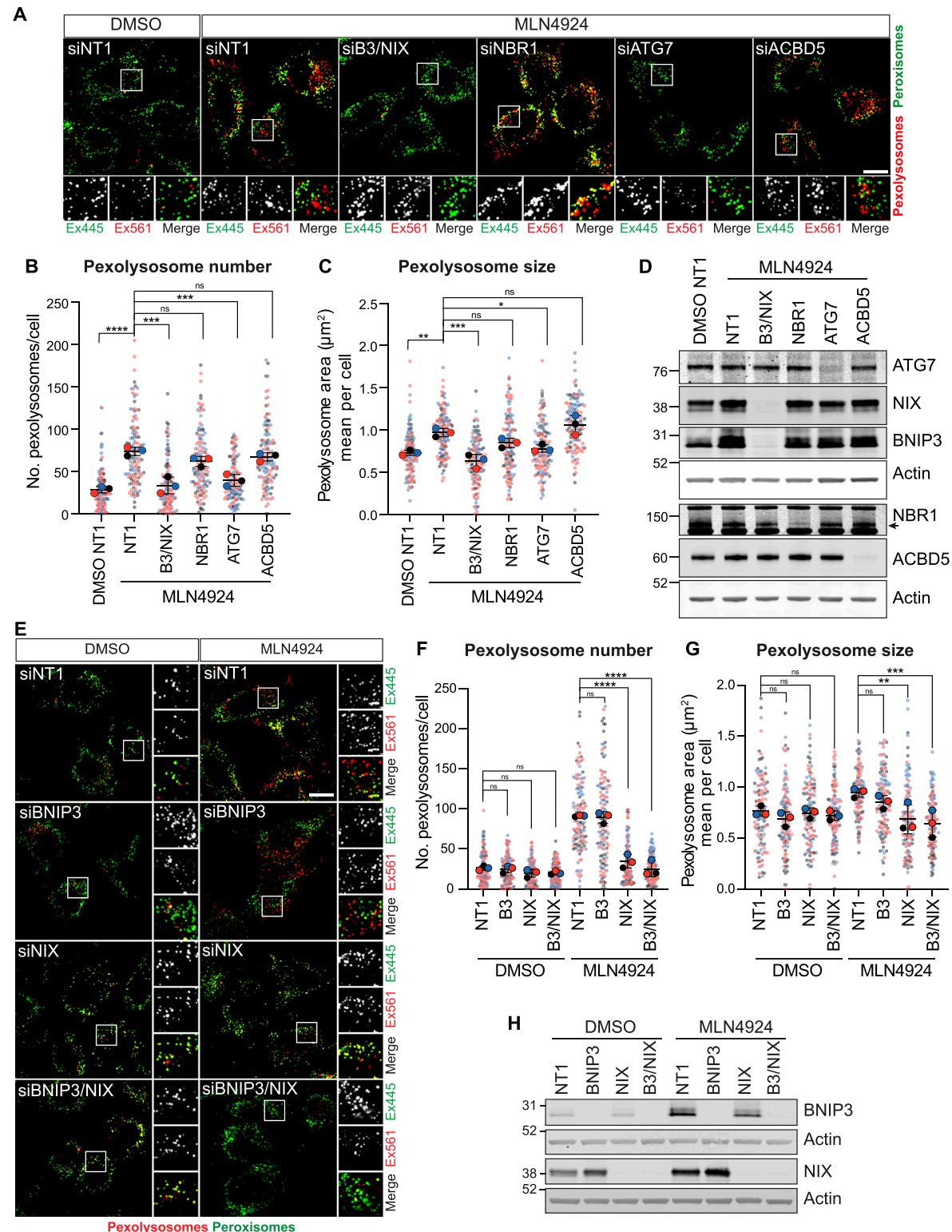

**Figure 2. MLN4924-induced pexophagy requires NIX.**

**(A)** Representative confocal images of hTERT-RPE1-Keima-SKL cells treated with DMSO and MLN4924 (1 µM) for 24 h before imaging. Cells were transfected with non-targeting siRNA (NT1) or siRNA targeting BNIP3, NIX, ATG7, ACBD5, and NBR1. Scale bar 20 µm. **(B, C)** Graphs show the number and mean area of pexolysosomes. Quantification of the data from three colour-coded independent experiments is shown. Mean and SD are indicated; >40 cells were quantified per condition in each experiment. One-way ANOVA with Bonferroni's multiple comparisons test. *$P < 0.05$. **$P < 0.01$. ***$P < 0.001$. ****$P < 0.0001$. **(D)** Representative Western blot of hTERT-RPE1-Keima-SKL cells treated as in (A) and probed as indicated. **(E)** Representative confocal images of hTERT-RPE1-Keima-SKL cells stably expressing the Keima-SKL pexophagy reporter. Cells were treated with DMSO and MLN4924 (1 µM), for 24 h before imaging. Cells were transfected with non-targeting (NT1) siRNA or siRNA targeting

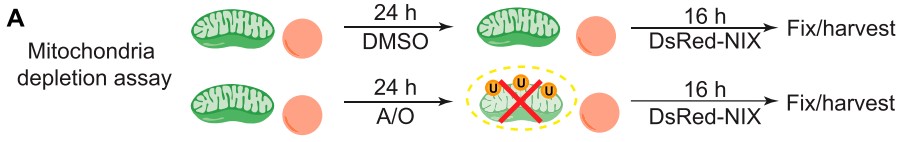

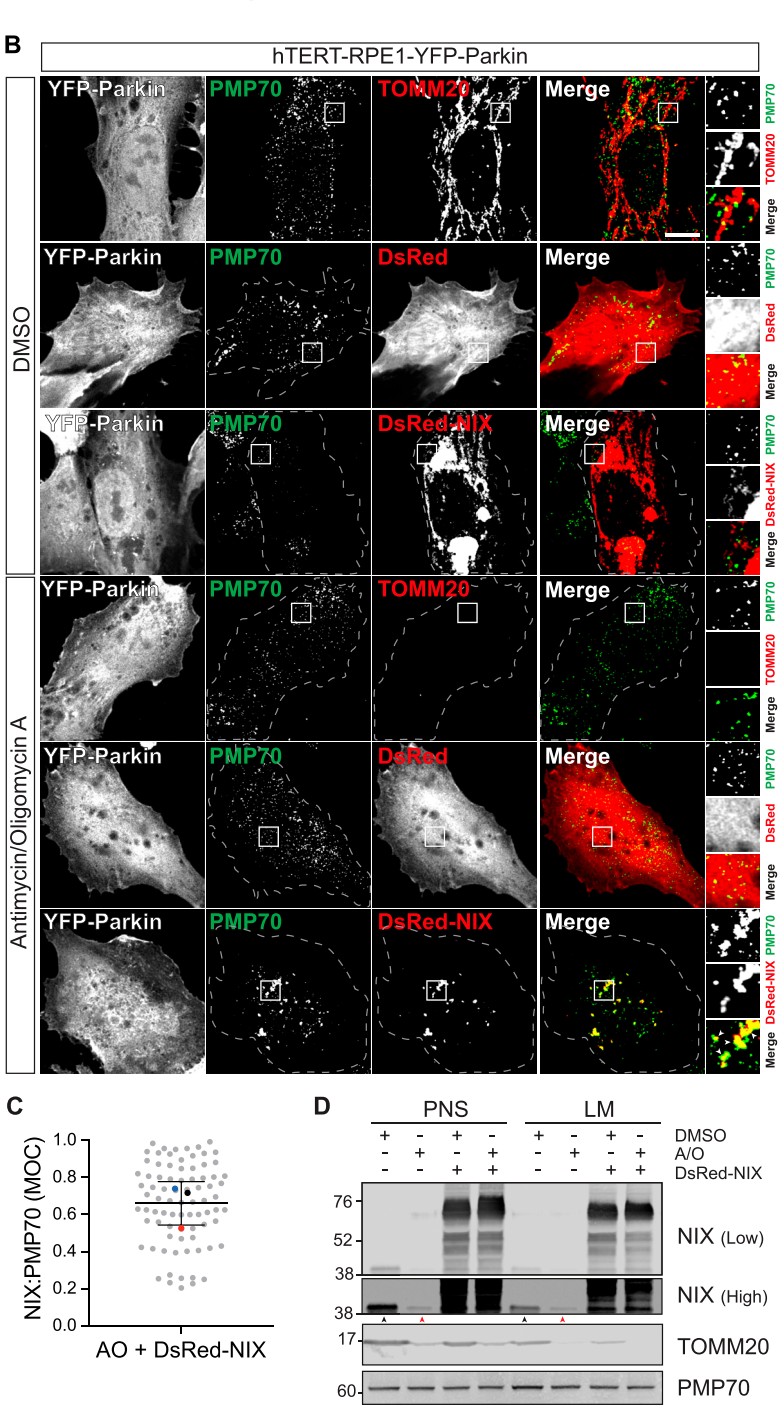

**Figure 3. Exogenously expressed NIX colocalises with peroxisomes independently of mitochondria.**
**(A)** hTERT-RPE1-YFP-Parkin cells were first treated for 24 h with antimycin A and oligomycin A (A/O, 1 $\mu$M each) or DMSO, then transiently transfected with DsRed-NIX or DsRed alone for 16 h before fixation or harvesting. **(B)** Representative images of cells, treated as described in (A), immunostained for endogenous PMP70 (AlexaFluor-405, green). A set of mock transfected cells treated in parallel were co-stained for TOMM20 (AlexaFluor-594, red). Scale bars 10 $\mu$m. **(C)** Colocalisation of the DsRed-NIX signal with peroxisomes (PMP70) in hTERT-RPE1-YFP-Parkin cells treated with A/O. The Mander's overlap coefficient (MOC, M1) was calculated for >25 cells per experiment. The mean and SD of three independent experiments are shown. **(D)** Representative Western blot showing subcellular fractions of hTERT-RPE1-YFP-Parkin cells treated as shown in (A). Postnuclear supernatant and light membranes are shown, representative of two independent experiments. Black and red arrows highlight relative amounts of endogenous NIX in the light membrane fraction, recovered from the postnuclear supernatant ± mitochondrial depletion (A/O treatment).

Source data are available for this figure.

BNIP3 and NIX. Scale bar 20 $\mu$m. **(F, G)** Graph shows the number and mean area of pexolysosomes per cell. Quantification of the data from three colour-coded independent experiments is shown. Mean and SD are indicated; >40 cells were quantified per condition in each repeat experiment. One-way ANOVA with Bonferroni's multiple comparisons test. **$P < 0.01. ***P < 0.001. ****P < 0.0001. **(H)** Representative Western blot of protein samples from cells treated as in (E) and probed as indicated. Source data are available for this figure.

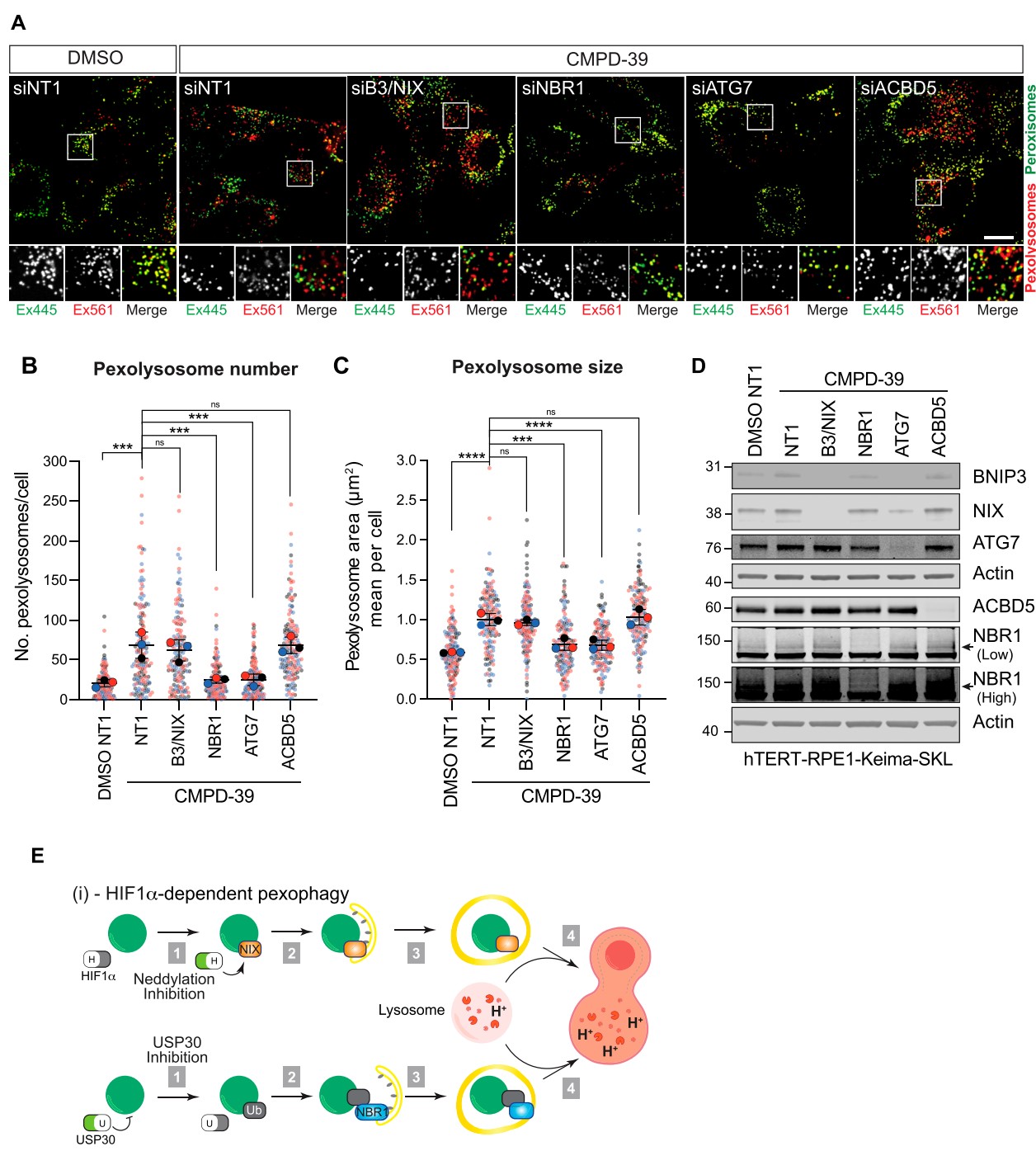

**Figure 4. NBR1 mediates CMPD-39–induced pexophagy.**
**(A)** Representative confocal images of hTERT-RPE1-Keima-SKL cells treated with DMSO and CMPD-39 (1 $\mu$M), for 96 h before imaging. Cells were transfected with non-targeting (NT1) or siRNA or siRNA targeting BNIP3, NIX, ATG7, ACBD5, and NBR1. Scale bar 20 $\mu$m. **(B, C)** Quantification of data shown in (A). Graphs show the number and mean area of pexolysosomes per cell from three independent colour-coded experiments. Mean and SD are indicated; >50 cells were quantified per condition in each experiment. One-way ANOVA with Bonferroni's multiple comparisons test. ***$P < 0.001$. ****$P < 0.0001$. **(D)** Representative Western blot of protein samples from cells treated as in (A) and probed as indicated. Low and high represent two different exposures of the same blot. **(E)** HIF1$\alpha$-dependent and -independent pexophagy pathways. (i) HIF1$\alpha$-dependent pexophagy pathway. Upon administration of the neddylation inhibitor MLN4924, the transcription factor HIF1$\alpha$ is induced (top). This leads to the up-regulation of NIX, which directly associates with peroxisomes and acts as a pexophagy adaptor. (ii) HIF1$\alpha$-independent pexophagy pathway: the DUB USP30 suppresses pexophagy by removing ubiquitin attached to peroxisomal substrates. Accumulation of ubiquitin on peroxisomes, after USP30 inhibition, leads to the recruitment of the NBR1 pexophagy adaptor.
Source data are available for this figure.

pexophagy (5, 37). However, we cannot exclude a supporting role for other ubiquitin-binding adaptors.

USP30 amplifies mitochondrial stress responses, which may be triggered by defective protein import or depolarisation (22, 38, 39, 40, 41, 42, 43, 44, 45). The simplest model predicts that peroxisome-localised USP30 suppresses critical ubiquitin signals, linked to quality control, that otherwise lead to pexophagy (22, 23). Based on our previous analysis of the ubiquitylome, after USP30 inhibition, we do not expect wholesale changes in the ubiquitylation profile of peroxisomes (44). USP30 has been proposed to counter the action of the peroxisomal E3-ligase PEX2 and suppress pexophagy in response to amino acid starvation (23). A recent study has linked accumulation of ubiquitylated PEX5 (a PEX2 substrate) and pexophagy to compromised peroxisomal import (46). Zheng and colleagues have shown that the E3-ligase MARCH5 is shared between mitochondria and peroxisomes and can also promote pexophagy (47). MARCH5 and USP30 at mitochondria have been shown to reciprocally regulate the ubiquitin status of the mitochondrial import (TOMM) complex (43). We have proposed that this sets a trigger threshold for unleashing the PINK1-PRKN cascade (22, 44, 48). We speculate that a MARCH5/USP30 counterpoise system may be preserved at the peroxisome and is disrupted by CMPD-39. Both pexophagy pathways, whose induction we have chemically segregated here, have key factors in common with established mitophagy pathways. Given the metabolic interplay between these two organelles, one can imagine the advantages of regulating their abundance in a coordinated manner.

# Materials and Methods

### Cell culture, transfection, and RNA interference

hTERT-RPE1, hTERT-RPE1-Keima-SKL, and hTERT-RPE1-YFP-Parkin cells were routinely cultured in Dulbecco's Modified Eagle's medium DMEM/F12 (31331028; Gibco) supplemented with 10% FBS (10270106; Gibco), 1% non-essential amino acids (111505035; Gibco), and 1% penicillin/streptomycin at 37°C and 5% $CO_2$. Cells were routinely checked for mycoplasma. For RNA interference experiments, cells were treated with 40 nM of non-targeting (NT1) or target-specific siRNA oligonucleotides (Dharmacon On-target plus smart pool), using Lipofectamine RNAi-MAX (13778030; Invitrogen) according to the manufacturer's instructions. For plasmid transfections, Lipofectamine 3000 (L3000001; Invitrogen) was used according to the manufacturer's instructions.

### Generation of the pexophagy reporter hTERT-RPE1-Keima-SKL cell line

hTERT-RPE1-Cas9i-PuroS cells (29 Preprint) were transfected with pCDNA3.1-mKeima-SKL-BlastR (derived from pCDNA3.1-mKeima-SKL-Neo (22)), using Lipofectamine 2000 (11668019; Invitrogen). Transfected cells were selected with 10 μg/ml blasticidin S HCl for 7 d and mKeima-positive cells were isolated by FACS. A clonally isolated cell line expressing suitable levels of the mKeima-SKL reporter was selected and is referred to as hTERT-RPE1-Keima-SKL for all experiments in this article.

### siRNA and plasmids

The following ON-Target Plus Smart Pool siRNAs were obtained from Dharmacon: BNIP3 (5′-UCGCAGACACCACAAGAUA-3′, 5′-GAACUGCA-CUUCAGCAAUA-3′, 5′-GGAAAGAAGUUGAAAGCAU-3′, 5′-ACACGAGCGU-CAUGAAGAA-3′), BNIP3L/NIX (5′-GACCAUAGCUCUCAGUCAG-3′, 5′-CAA CAACAACUGCGAGGAA-3′, 5′-GAAGGAAGUCGAGGCUUUG-3′, 5′-GAGAAU UGUUUCAGAGUUA-3′), ATG7 (5′-CCAACACACUCGAGUCUUU-3′, 5′-GAUC UAAAUCUCAAACUGA-3′, 5′-GCCCACAGAUGGAGUAGCA-3′, 5′-GCCAGAG GAUUCAACAUGA-3′), ACBD5 (5′-CUAAAGGGAUCUACUACUA-3′, 5′-CCA AAACCGUUAAUGGUAA-3′, 5′-CAGCAUUUGACAAGCGAUU-3′, 5′-GGAU GCAACACUUGAGCGA-3′), NBR1 (5′-GAGAACAAGUGGUUAACGA-3′, 5′-CC ACAUGACAGUCCUUUAA-3′, 5′-GAACGUAUACUUCCCAUUG-3′, 5′-AGAA GCCACUUGCACAUUA-3′). DsRed-BNIP3L/NIX plasmid was obtained from Addgene (100763). DsRed-N1 plasmid was a gift from Francis Barr (University of Oxford).

### Antibodies and reagents

Antibodies and other reagents used were as follows: anti-BNIP3L (#12396, 1:1,000 WB, 1:250 IF; Cell Signalling), anti-BNIP3 (ab109362, 1:1,000 WB; Abcam), anti-TOM20 (11802-1-AP, 1:1,000 WB; ProteinTech), anti-TOM20 (HPA011562, 1:500 IF; Sigma-Aldrich), anti-PMP70 (SAB4200181, 1:1,000 WB, 1:250 IF; Sigma-Aldrich), mouse anti-actin (66009-1-Ig, 1:10,000; ProteinTech), anti-Cullin-2 (A302-476A, 1:2,000 WB; Bethyl Laboratories), anti-HIF1α (NB100-134, 1:1,000; Novus Bio techne), anti-ATG7 (2631, 1:1,000 WB; Cell Signalling), anti-NBR1 (9891S, 1:1,000 WB; Cell Signalling), anti-LC3 (5F10, 1:200 WB; Nanotools), MLN4924 (C-1231; Chemgood), deferiprone (379409; Sigma-Aldrich), 4-PBA (21005; Sigma-Aldrich), clofibrate (C6643; Sigma-Aldrich), hydrogen peroxide (H1009; Sigma-Aldrich), oligomycin A (75351; Sigma-Aldrich), antimycin A (A8674; Sigma-Aldrich), Mitotracker Deep Red FM (M22426; Invitrogen).

### Preparation of cell lysates and Western blot analysis

Cultured cells were lysed with RIPA buffer (10 mM Tris–HCl pH 7.5, 150 mM sodium chloride, 1% sodium deoxycholate, 0.1% SDS, 1% Triton X-100) and routinely supplemented with mammalian protease inhibitor cocktail (P8340; Sigma-Aldrich) and PhosSTOP (04906837001; Roche). Proteins were resolved using SDS–PAGE (NuPage gel 4–12%, NP0321/NP0335; Invitrogen), transferred to the nitrocellulose membrane (10600002; Amersham), blocked in 5% milk (Marvel) or 5% BSA (First Link, 41-10-410) in TBS (20 mM Tris-Cl, pH 7.6, 150 mM NaCl), supplemented with Tween-20 (10485733; Thermo Fisher Scientific), and probed with primary antibodies overnight. Visualization and quantification of Western blots were performed using IRdye 800CW and 680LT-coupled secondary antibodies and an Odyssey infrared scanner (LI-COR Biosciences).

### Mitochondria depletion assay

As previously described (22), hTERT-RPE1 cells stably overexpressing YFP-Parkin (hTERT-RPE1-YFP-Parkin) were treated for 24 h with antimycin A (1 μM) and oligomycin A (1 μM), washed with PBS, and cultured in normal media before transfection with a plasmid expressing DsRED alone or DsRed-BNIP3L/NIX (22). After

16 h, cells were fixed for immunofluorescence microscopy or harvested for subcellular fractionation.

## Subcellular fractionation

hTERT-RPE1-YFP-Parkin cells were washed twice with ice-cold PBS, followed by centrifugation at 1,000$g$ for 2 min at 4°C. Cell pellets were resuspended in HIM buffer (200 mM mannitol, 70 mM sucrose, 1 mM EGTA, and 10 mM Hepes-NaOH, pH 7.4) and centrifuged again at 1,000$g$ for 5 min at 4°C. The cell pellet was resuspended in HIM buffer, supplemented with 50 mM 2-chloroacetamide, mammalian protease inhibitor cocktail (P8340; Sigma-Aldrich), and PhosSTOP (04906837001; Roche), before cells were mechanically disrupted by shearing with a 23-gauge needle. The cell homogenate was then centrifuged at 600$g$ for 10 min to obtain the postnuclear supernatant. Heavy membranes enriched in mitochondria were removed by centrifugation at 7,000$g$ for 15 min, and the supernatant was centrifuged at 100,000$g$ for 30 min to generate the light membrane pellet fraction (LM). Equal amount of protein from both postnuclear supernatant and LM (10 $\mu$g/lane) was resolved by SDS–PAGE (NuPage gel 4–12%; Invitrogen).

## Immunofluorescence and colocalization analysis

Cells were fixed using 4% paraformaldehyde in PBS, permeabilised with 0.2% Triton X-100 in PBS, stained with AlexaFluor-405, -488, or -594–coupled secondary antibodies, and imaged using a Zeiss LSM900 with Airyscan (63× NA 1.4 oil, acquisition software Zen Blue). The images were processed using Adobe Photoshop 2022 and Fiji v2.9.0 software. For colocalisation analysis, single confocal z-planes were analysed with the JaCoP plugin in Fiji v2.9.0 to derive the Mander's overlap coefficient M1. Quantification of colocalisation was performed from three independent experiments analysing >25 cells per experiment.

## Live-cell imaging of pexophagy

For live cell imaging, hTERT-RPE1-Keima-SKL cells were seeded onto an IBIDI $\mu$-Dish (2 × $10^5$) (81156; IBIDI) 2 d before image acquisition using a 3i Marianas spinning disk confocal microscope (63× oil objective, NA 1.4, Photometrics Evolve EMCCD camera, Slide Book 3i v3.0). Live cells were imaged sequentially (Ex445/Em600 then Ex561/Em600). Images were processed using Adobe Photoshop 2022 and Fiji v2.9.0 softwares. Analysis of pexophagy levels in hTERT-RPE1-Keima-SKL was performed using the semi-automated "mito-QC Counter" plugin implemented in Fiji v2.9.0 software as previously described (49). The analysis of pexophagy was performed for three independent experiments analysing >40 cells per condition in each experiment.

## Statistical analysis

$P$-values are indicated as *$P$ < 0.05, **$P$ < 0.01, ***$P$ < 0.001, ****$P$ < 0.0001 and derived by one-way ANOVA and Bonferroni's multiple comparisons post hoc test. All statistical analyses were conducted using GraphPad Prism 9.

# Supplementary Information

# Acknowledgements

hTERT-RPE1-YFP-Parkin cells were a kind gift of Jon Lane (University of Bristol). FG Barone was funded by a Wellcome Trust PhD studentship, 102172/B/13/Z. MJ Clague is a Royal Society Industry Fellow, INF\R2\212031. Additional support was provided by Bristol-Myers Squibb. We are also grateful to the Liverpool University Centre for Cell Imaging for access to instrumentation.

## Author Contributions

FG Barone: conceptualization, data curation, formal analysis, validation, investigation, visualization, methodology, and writing—original draft, review, and editing.
S Urbé: conceptualization, resources, data curation, formal analysis, supervision, funding acquisition, visualization, methodology, project administration, and writing—original draft, review, and editing.
MJ Clague: conceptualization, data curation, supervision, funding acquisition, methodology, project administration, and writing—original draft, review, and editing.

## Conflict of Interest Statement

The authors declare that they have no conflict of interest.

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
