## [Reviewer comments · Life Science Alliance]

Segregation of pathways leading to pexophagy

Francesco Barone, Sylvie Urbe, and Michael Clague

DOI: <https://doi.org/10.26508/lsa.202201825>

Corresponding author(s): Michael Clague, University of Liverpool

Review Timeline:

Submission Date:	2022-11-14
Editorial Decision:	2022-12-15
Revision Received:	2023-01-26
Editorial Decision:	2023-02-10
Revision Received:	2023-02-12
Accepted:	2023-02-13

Transaction Report:

December 15, 2022

Re: Life Science Alliance manuscript #LSA-2022-01825

Prof. Michael J. Clague
University of Liverpool
Cellular and Molecular Physiology, Biomedical Sciences
Crown St.
Liverpool, Merseyside L69 3BX
United Kingdom

Dear Dr. Clague,

Thank you for submitting your manuscript entitled "Segregation of pathways leading to pexophagy" to Life Science Alliance. The manuscript was assessed by expert reviewers, whose comments are appended to this letter. We invite you to submit a revised manuscript addressing the Reviewer comments.

Thank you for this interesting contribution to Life Science Alliance. We are looking forward to receiving your revised manuscript.

Sincerely,

B. MANUSCRIPT ORGANIZATION AND FORMATTING:

Reviewer #1 (Comments to the Authors (Required)):

Pexophagy is a type of macro-autophagy that selectively degrades peroxisomes. The manuscript by Barone et al. investigates drugs and pathways linked with pexophagy, with special focus on the role of Hif-induced NIX in pexophagy. There are some novel findings presented which will improve our knowledge about this underexplored autophagy mechanism. Overall, the paper is well written and of interest to the wide autophagy community.

Initially the authors test a small set of drugs/compounds known to affect pexophagy and/or mitophagy. This demonstrated that CMPD-39 (USP30 inhibitor) and MLN4924 (neddylation inhibitor) are potent inducers of pexophagy. Subsequently the authors demonstrated that MLN4924-induced pexophagy requires canonical autophagy/ATG7 and NIX. The authors then show that over-expressed NIX colocalise to peroxisomes in cells depleted of mitochondria (although additional work would strengthen this part). Lastly, the authors show that pexophagy induced by CMPD-39 depends on canonical autophagy/ATG7 and NBR1 (but not NIX) suggesting it requires LC3-NBR1-ubiquitin interaction and therefore occurs through separate pathway compared to MLN4924-induced pexophagy.

Taken together, using a relevant in vitro model of pexophagy induced upon chemical stabilization of Hif1 α , the authors present novel data suggesting the involvement of NIX in the removal of peroxisomes via direct localization of this protein into the organelles undergoing degradation. Moreover, the authors suggest the existence of a separate NIX-independent pexophagy mechanism, induced by CMPD-39 (suggested to be ubiquitin dependent). The manuscript sheds light on the idea that NIX can induce removal of both mitochondria and peroxisomes (and is in line with recently published works). However, some points would need addressing to improve the quality/potential impact of this manuscript (see specific comments below).

Specific comments:

Figure 1: It would be informative to visualise the changes in LC3 lipidation following the drug treatments.

Can the authors compare the effect of DFP treatment with hypoxia on NIX-dependent pexophagy (i.e. test/show that hypoxia-induced pexophagy also requires ATG7 and NIX, but not NBR1)?

Figure 3D: Here it seems that the overexpressed NIX localises into peroxisomes. However, it would be nice to see that this happens with endogenous NIX to make sure it is not an "artefact" of its overexpression. Additionally, if co-treatment with A/O is promoting NIX's migration to peroxisomes following CMPD-39 treatment in mitochondrial depleted cells, how does this translocation compare with the translocation in mitochondrial containing cells (i.e., is more endogenous NIX available for peroxisome localisation when mitochondria has been removed)?

Can the authors visualise ubiquitination on peroxisomes following CMPD-39 treatment?

Can the authors also discuss/speculate about the physiological role for pexophagy. What are possible biological differences between the two different pathways suggested? Under what conditions (peroxisome damage?) would the separate mechanism likely take place?

Minor Comment:

Can the authors change or clarify what they mean with the sentence: "Whilst NIX expression is under the control of ubiquitin E-3-ligase activity"?

Reviewer #2 (Comments to the Authors (Required)):

In the manuscript by Barone, Urbe and Clague entitled "Segregation of pathways leading to pexophagy" the authors show that

the neddylation inhibitor, MLN4924, actually acts as a potent activator of pexophagy. The authors demonstrate that this is due to HIF1 α -dependent upregulation of BNIP3L/NIX. NIX is a well known mitophagy receptor upregulated by hypoxia and acting in the removal of mitochondria during reticulocyte development and neuronal differentiation. The authors study here autophagic degradation of peroxisomes (pexophagy) using human retinal pigment epithelial cells, hTERT-RPE1, expressing a tetracycline-inducible pexophagy reporter consisting of mKeima fused to the C-terminal peroxisomal protein matrix targeting signal SKL to report pexophagy by fluorescent confocal microscopy. They study several chemicals to determine their relative abilities to induce pexophagy. These are 4-phenylbutyric acid - 4-PBA, Clofibrate, Deferiprone - DFP, MLN4924, and the USP30 inhibitor CMPD-39. All of these agents induced pexophagy and the authors then focused on the most specific acting chemicals, MLN4924 inhibiting the neddylation dependent activation of Cullins as well as the mentioned USP30 inhibitor CMPD-39. Through very nicely performed experiments the authors find that these two drugs induce two different pexophagy pathways. MLN4924 leads to elevated levels of the mitophagy receptor NIX and NIX is shown to also be located on the membrane of peroxisomes and to induce pexophagy.

The dual role of NIX as both a mitophagy and pexophagy receptor reported here is completely consistent with a study recently published by the group of Ian Ganley in EMBO J (DOI 10.15252/embj.2022111115) while this study was in progress. The study in EMBO J used DFP to raise NIX levels and not MLN4924 as done here. Barone et al go on to show that CMPD-39 actually induces a separate pathway of pexophagy requiring NBR1 as the pexophagy receptor.

It is an interesting study pointing to the diversity in pexophagy pathways. Pexophagy is not well studied compared to mitophagy making this study a wellcome one. The authors also describe the use of these chemicals that can be used to induce two different pexophagy pathways. These will be nice tools for future studies.

The paper is exemplary written, the experiments are very well performed and the data are presented in an excellent manner. I will like to give flowers to the authors for their nice visualization of their quantification of three different experiments the way they have done here.

In my view this work is acceptable for publication in LSA as is.

Reviewer #3 (Comments to the Authors (Required)):

A straight-forward, well-performed and evocative study suggesting a new level of interplay between the machineries controlling mitophagy and pexophagy. Whilst the scope of the study is limited and the mechanism of this coordinated control of pexophagy with mitophagy is not fully resolved, the authors present intriguing and generally compelling data to delineate the pexophagy pathways mediated by Nix and Nbr1. As noted by the authors, aspects of this study are consistent with Wilhelm et al who recently reported a role for BNIP3/Nix in pexophagy induced by iron chelation with deferiprone. Whilst this impacts on the conceptual advance, I think it is important to publish timely corollary findings.

1. The authors conclude that the Nix upregulation is Hif1 α -dependent. Whilst Hif1 α upregulation clearly correlates, to confirm dependence the authors should deplete Hif1 α and confirm Nix upregulation is abrogated.
2. Fig 3C- Present the colocalization coefficient data for Nix and Tom20 in untreated cells for comparison.
3. Fig 3D- confirm subcellular fractionation with endogenous Nix by providing a longer exposure of the existing blot, or modify gel running to better reveal endogenous signal.
4. Pexophagy in response to CMPD39 was reduced following siNBR1, but it was not completely blocked even in individual cells where pexophagy was clearly impaired (at least based on the images in Fig 4A). This was presumably due to incomplete si-mediated knockdown of NBR1, or alternatively redundancy in the pathway. Suggest Crispr-deletion to confirm reliance on NBR1 or show that other autophagy adaptors are not involved.
5. If possible, incorporating the known roles for these proteins in mitophagy into the schematic in 4E to inform the coordinated control would be useful for the general reader.

We thank the collective efforts of all reviewers. Our responses to individual comments are detailed below.

Reviewer #1 (Comments to the Authors (Required)):

Pexophagy is a type of macro-autophagy that selectively degrades peroxisomes. The manuscript by Barone et al. investigates drugs and pathways linked with pexophagy, with special focus on the role of Hif-induced NIX in pexophagy. There are some novel findings presented which will improve our knowledge about this underexplored autophagy mechanism. Overall, the paper is well written and of interest to the wide autophagy community.

Initially the authors test a small set of drugs/compounds known to affect pexophagy and/or mitophagy. This demonstrated that CMPD-39 (USP30 inhibitor) and MLN4924 (neddylation inhibitor) are potent inducers of pexophagy. Subsequently the authors demonstrated that MLN4924-induced pexophagy requires canonical autophagy/ATG7 and NIX. The authors then show that over-expressed NIX colocalise to peroxisomes in cells depleted of mitochondria (although additional work would strengthen this part). Lastly, the authors show that pexophagy induced by CMPD-39 depends on canonical autophagy/ATG7 and NBR1 (but not NIX) suggesting it requires LC3-NBR1-ubiquitin interaction and therefore occurs through separate pathway compared to MLN4924-induced pexophagy.

Taken together, using a relevant in vitro model of pexophagy induced upon chemical stabilization of Hif1 α , the authors present novel data suggesting the involvement of NIX in the removal of peroxisomes via direct localization of this protein into the organelles undergoing degradation. Moreover, the authors suggest the existence of a separate NIX-independent pexophagy mechanism, induced by CMPD-39 (suggested to be ubiquitin dependent). The manuscript sheds light on the idea that NIX can induce removal of both mitochondria and peroxisomes (and is in line with recently published works). However, some points would need addressing to improve the quality/potential impact of this manuscript (see specific comments below).

We thank the reviewer for their positive comments and nice summary of our findings.

Specific comments:

Figure 1: It would be informative to visualise the changes in LC3 lipidation following the drug treatments.

Yes we have now added this blot to Figure 1E.

Can the authors compare the effect of DFP treatment with hypoxia on NIX-dependent pexophagy (i.e, test/show that hypoxia-induced pexophagy also requires ATG7 and NIX, but not NBR1)?

Preceding work of the Ganley laboratory which we reference has compared DFP and hypoxia with both showing NIX-dependence. The focus of our paper is on MLN4924 which we have shown to elevate NIX by inhibition of VHL (elevates HIF levels and transcription) and FBXL4 (reduces turnover). In fact both DFP and MLN2924 mimic hypoxia.

Figure 3D: Here it seems that the overexpressed NIX localises into peroxisomes. However, it would be nice to see that this happens with endogenous NIX to make sure it is not an "artefact" of its overexpression. Additionally, if co-treatment with A/O is promoting NIX's migration to peroxisomes following CMPD-39 treatment in mitochondrial depleted cells, how does this translocation compare with the translocation in mitochondrial containing cells (i.e., is more endogenous NIX available for peroxisome localisation when mitochondria has been removed)?

Reference 33 shows that DFP induced endogenous NIX localises to peroxisomes (Fig EV3A). There is perhaps some confusion to the second part of this question, we do not make any claims that NIX migrates to peroxisomes following CMPD-39 treatment. We have now added a higher exposure NIX blot to Figure 3D that allows visualisation of the endogenous form. A loss of NIX in the PNS and light membrane fractions reflects the loss of mitochondria. In this residual setting (after AO) a much higher proportion of endogenous NIX, in fact nearly all, is now recovered in the light membrane fraction, consistent with peroxisomal localisation.

Can the authors visualise ubiquitination on peroxisomes following CMPD-39 treatment?

We have previously performed global ubiquitinomics on USP30 inhibitor treated cells and not observed major shifts in the ubiquitin status of peroxisomal proteins or in fact mitochondrial proteins i.e. the total ubiquitin mass associated with each organelle (ref. 44). We have now explicitly stated this in the discussion section.

Can the authors also discuss/speculate about the physiological role for pexophagy. What are possible biological differences between the two different pathways suggested? Under what conditions (peroxisome damage?) would the separate mechanism likely take place?

We have modified the discussion section to reflect this point; proposing roles in metabolic reprogramming (MLN4924/hypoxia) and peroxisomal quality control (CMPD39).

Minor Comment:

Can the authors change or clarify what they mean with the sentence: "Whilst NIX expression is under the control of ubiquitin E-3-ligase activity"?

Yes this refers to the dual control of NIX by the Cullin associated proteins VHL and FBXL4. We have expanded on this and added the relevant reference (29).

Reviewer #2 (Comments to the Authors (Required)):

In the manuscript by Barone, Urbe and Clague entitled "Segregation of pathways leading to pexophagy" the authors show that the neddylation inhibitor, MLN4924, actually acts as a potent activator of pexophagy. The authors demonstrate that this is due to HIF1 α -dependent upregulation of BNIP3L/NIX. NIX is a well known mitophagy receptor upregulated

by hypoxia and acting in the removal of mitochondria during reticulocyte development and neuronal differentiation. The authors study here autophagic degradation of peroxisomes (pexophagy) using human retinal pigment epithelial cells, hTERT-RPE1, expressing a tetracycline-inducible pexophagy reporter consisting of mKeima fused to the C-terminal peroxisomal protein matrix targeting signal SKL to report pexophagy by fluorescent confocal microscopy. They study several chemicals to determine their relative abilities to induce pexophagy. These are 4-phenylbutyric acid - 4-PBA,

Clofibrate, Deferiprone - DFP, MLN4924, and the USP30 inhibitor CMPD-39. All of these agents induced pexophagy and the authors then focused on the most specific acting chemicals, MLN4924 inhibiting the neddylation dependent activation of Cullins as well as the mentioned USP30 inhibitor CMPD-39. Through very nicely performed experiments the authors find that these two drugs induce two different pexophagy pathways. MLN4924 leads to elevated levels of the mitophagy receptor NIX and NIX is shown to also be located on the membrane of peroxisomes and to induce pexophagy. The dual role of NIX as both a mitophagy and pexophagy receptor reported here is completely consistent with a study recently published by the group of Ian Ganley in EMBO J (DOI 10.15252/embj.2022111115) while this study was in progress. The study in EMBO J used DFP to raise NIX levels and not MLN4924 as done here. Barone et al go on to show that CMPD-39 actually induces a separate pathway of pexophagy requiring NBR1 as the pexophagy receptor.

It is an interesting study pointing to the diversity in pexophagy pathways. Pexophagy is not well studied compared to mitophagy making this study a wellcome one. The authors also describe the use of these chemicals that can be used to induce two different pexophagy pathways. These will be nice tools for future studies.

The paper is exemplary written, the experiments are very well performed and the data are presented in an excellent manner. I will like to give flowers to the authors for their nice visualization of their quantification of three different experiments the way they have done here.

In my view this work is acceptable for publication in LSA as is.

We warmly appreciate these comments from the referee.

Reviewer #3 (Comments to the Authors (Required)):

A straight-forward, well-performed and evocative study suggesting a new level of interplay between the machineries controlling mitophagy and pexophagy. Whilst the scope of the study is limited and the mechanism of this coordinated control of pexophagy with mitophagy is not fully resolved, the authors present intriguing and generally compelling data to delineate the pexophagy pathways mediated by Nix and Nbr1. As noted by the authors, aspects of this study are consistent with Wilhelm et al who recently reported a role for BNIP3/Nix in pexophagy induced by iron chelation with deferiprone. Whilst this impacts on the conceptual advance, I think it is important to publish timely corollary findings.

We thank the reviewer for this accurate and supportive account of our work.

1. The authors conclude that the Nix upregulation is Hif1a-dependent. Whilst Hif1a upregulation clearly correlates, to confirm dependence the authors should deplete Hif1a and confirm Nix upregulation is abrogated.

We think its a little bit more complicated than that, MLN4924 is acting not only on VHL/HIF but also FBXL4, which destabilises NIX. We have expanded on this point in the text in response to reviewer 1 citing reference 29, where the experiment the referee requests is reported.

2.Fig 3C- Present the colocalization coefficient data for Nix and Tom20 in untreated cells for comparison.

Figure 3C shows co-localisation of NIX with a peroxisomal marker. It is well established that NIX localises to mitochondria, the novel point of our manuscript is the peroxisomal localisation. We think the referee is asking to see NIX on mitochondria in the absence of mitochondrial clearance by AO. For this we have to use the parental cells, which do not express YFP-Parkin. We now present a further supplementary Figure S2 which shows endogenous NIX localisation to mitochondria using a mitotracker dye that is well retained after fixation. We have used pre-treatment with MLN4924 to increase NIX expression and amplify the NIX fluorescence signal.

3.Fig 3D- confirm subcellular fractionation with endogenous Nix by providing a longer exposure of the existing blot, or modify gel running to better reveal endogenous signal.

Yes we have done this, see response to reviewer 1.

4.Pexophagy in response to CMPD39 was reduced following siNBR1, but it was not completely blocked even in individual cells where pexophagy was clearly impaired (at least based on the images in Fig 4A). This was presumably due to incomplete si-mediated knockdown of NBR1, or alternatively redundancy in the pathway. Suggest

Crispr-deletion to confirm reliance on NBR1 or show that other autophagy adaptors are not involved.

We now explicitly state that we cannot exclude a minor involvement of other ubiquitin binding adaptors. Our claim here is that NBR1 can account for the major component of CMPD-39 induced pexophagy and that crucially this pathway does not involve NIX or BNIP3L.

5. If possible, incorporating the known roles for these proteins in mitophagy into the schematic in 4E to inform the coordinated control would be useful for the general reader.

We understand this suggestion and have considered it carefully, however we think this would make the image too busy and detract from the clarity of messaging.

February 10, 2023

RE: Life Science Alliance Manuscript #LSA-2022-01825R

Prof. Michael J. Clague
University of Liverpool
Cellular and Molecular Physiology, Biomedical Sciences
Crown St.
Liverpool, Merseyside L69 3BX
United Kingdom

Dear Dr. Clague,

Thank you for submitting your revised manuscript entitled "Segregation of pathways leading to pexophagy". We would be happy to publish your paper in Life Science Alliance pending final revisions necessary to meet our formatting guidelines.

- please add the author contributions and a conflict of interest statement to the main manuscript text
- please use the [10 author names, et al.] format in your references (i.e. limit the author names to the first 10)

A. FINAL FILES:

B. MANUSCRIPT ORGANIZATION AND FORMATTING:

****It is Life Science Alliance policy that if requested, original data images must be made available to the editors. Failure to provide original images upon request will result in unavoidable delays in publication. Please ensure that you have access to all original**

data images prior to final submission.**

The license to publish form must be signed before your manuscript can be sent to production. A link to the electronic license to publish form will be sent to the corresponding author only. Please take a moment to check your funder requirements.

Sincerely,

Reviewer #1 (Comments to the Authors (Required)):

The authors have addressed all my concerns and I have no further comments.

Reviewer #3 (Comments to the Authors (Required)):

The authors have adequately responded to/addressed my comments.

February 13, 2023

RE: Life Science Alliance Manuscript #LSA-2022-01825RR

Prof. Michael J. Clague
University of Liverpool
Cellular and Molecular Physiology, Biomedical Sciences
Crown St.
Liverpool, Merseyside L69 3BX
United Kingdom

Dear Dr. Clague,

Thank you for submitting your Research Article entitled "Segregation of pathways leading to pexophagy". It is a pleasure to let you know that your manuscript is now accepted for publication in Life Science Alliance. Congratulations on this interesting work.

DISTRIBUTION OF MATERIALS:

Again, congratulations on a very nice paper. I hope you found the review process to be constructive and are pleased with how the manuscript was handled editorially. We look forward to future exciting submissions from your lab.

Sincerely,
